Population structure of wild soybean (Glycine soja) based on SLAF-seq have implications for its conservation

Meng Jing 1
Yang Guoqian 2
Li Xuejiao 1
Zhao Yan 1
He Shuilian 1 heshuilian2006@163.com
1 College of Horticulture and Landscape, Yunnan Agricultural University , Kunming, Yunnan , China
2 Kunming Institute of Botany, Chinese Academy of Sciences , Kunming, Yuannan , China
Shrestha Jiban
Electronic publication date: 2023 Nov 8
Publication date: 2023
Volume: 11
Electronic Location ID: e16415
Received 2023 Aug 2; Accepted 2023 Oct 17
Copyright: © 2023 Meng et al.
Copyright year: 2023
Copyright holder: Meng et al.
License: This is an open access article distributed under the terms of the Creative Commons Attribution License, which permits unrestricted use, distribution, reproduction and adaptation in any medium and for any purpose provided that it is properly attributed. For attribution, the original author(s), title, publication source (PeerJ) and either DOI or URL of the article must be cited.
License URL: https://creativecommons.org/licenses/by/4.0/

Keywords: Glycine soja, SLAF-seq, Genetic diversity, Population structure

Funding: National Natural Science Foundation of China 32060083 & 31500459 Grant of Young Talents of Yunnan Xingdian Talents The project was supported by the National Natural Science Foundation of China (Project numbers 32060083 & 31500459); and the Grant of Young Talents of Yunnan Xingdian Talents. The funders had no role in study design, data collection and analysis, decision to publish, or preparation of the manuscript.

==============================
Background

Glycine soja Sieb. & Zucc. is the wild ancestor from which the important crop plant soybean was bred. G. soja provides important germplasm resources for the breeding and improvement of cultivated soybean crops, however the species is threatened by habitat loss and fragmentation, and is experiencing population declines across its natural range. Understanding the patterns of genetic diversity in G. soja populations can help to inform conservation practices.

Methods

In this study, we analyzed the genetic diversity and differentiation of G. soja at different sites and investigated the gene flow within the species. We obtained 147 G. soja accessions collected from 16 locations across the natural range of the species from China, Korea and Japan. Samples were analyzed using SLAF-seq (Specific-Locus Amplified Fragment Sequencing).

Results

We obtained a total of 56,489 highly consistent SNPs. Our results suggested that G. soja harbors relatively high diversity and that populations of this species are highly differentiated. The populations harboring high genetic diversity, especially KR, should be considered first when devising conservation plans for the protection of G. soja, and in situ protection should be adopted in KR. G. soja populations from the Yangtze River, the Korean peninsula and northeastern China have a close relationship, although these areas are geographically disconnected. Other populations from north China clustered together. Analysis of gene flow suggested that historical migrations of G. soja may have occurred from the south northwards across the East-Asia land-bridge, but not across north China. All G. soja populations could be divided into one of two lineages, and these two lineages should be treated separately when formulating protection policies.

Introduction

Portions of this text were previously published as part of a preprint (https://www.researchsquare.com/article/rs-2580996/v1).

Glycine soja Sieb. & Zucc., the wild soybean, is the ancestor from which the important crop plant soybean was bred (Smil, 2000). G. soja has a wide distribution throughout the Sino-Japanese Floristic Region (SJFR), between 24° and 53° N, and between 97° and 143° E. The species grows as a weed in cultivated land, on banks and in wetlands, from sea level to altitudes of 2,650 m (Lu, 2004). Outcrossing rates are thought to range from 2.4% to 19% (Kiang, Chiang & Kaizuma, 1992; Fujita et al., 1997) and the mean outcrossing rate of 77 G. soja populations in Japan was estimated to be 3.4% (Kuroda et al., 2006). The mean seed dispersal distances are only 10 m (Jin, He & Lu, 2003), and short distance seed dispersal is thought to occur mainly through pod dehiscence (Oka, 1983), while longer dispersal may be mediated by water or birds (Kiang, Chiang & Kaizuma, 1992; Choi et al., 1999; Kuroda et al., 2006). G. soja is distributed naturally in open habitats, which are often subject to human disturbance, and its distribution has therefore been significantly fragmented and reduced by human land exploitation and utilization. This species is even extinct in the wild in some regions and has been listed as a rare and endangered plant in China (Li, 1993). Genetic diversity is important to allowing a species to adapt to a changing environment and survive (Frankham, 2005), and elucidate the mechanisms underlying the origin and maintenance of genetic diversity is a fundamental task in biology (Mayr, 1963). Detailed knowledge of genetic variation can be applied to reveal the population structure and demographic history of a species (Novembre & Stephens, 2008) and to guide the formulation of conservation strategies for endangered species (Coop et al., 2010).

The evolutionary relationships between different G. soja populations have been investigated in the past mainly through the study of isozymes, DNA loci, SSRs and morphological characters (Dong et al., 2001; Li & Nelson, 2002; Zhao et al., 2005, 2018; Wang & Takahata, 2007; Wang, Li & Li, 2008; Li, Wang & Jia, 2009; Zhao, Xia & Lu, 2009; Lee et al., 2010; Wang et al., 2010; Wang & Li, 2011; He et al., 2012; Wang, Li & Liu, 2012; Wang, Li & Yan, 2014; Nawaz et al., 2017). Several molecular marker-based studies have discussed phylogeographic issues including geographical origins and patterns of dispersal (Choi et al., 1999; Kuroda et al., 2006, 2008, 2010; He et al., 2012), and one previous study used nuclear microsatellites and a chloroplast locus in combination with ecological niche modeling in a multidisciplinary approach to investigate the demographic history of G. soja (He et al., 2016). The distribution of G. soja during the LGM was found to be limited to southern and central China, and the species may have experienced extensive range expansion into northern East Asia following the end of the LGM. However, the genetic diversity of G. soja in northeastern China is very high. It is not clear whether marker selection is insufficient or whether the species has experienced rapid radiation or mutation. The limited number of polymorphic microsatellite sites used in this study did not result in good resolution of the soybean populations.

Study of the genetic diversity and population genetics of species can be conducted at different DNA molecular markers, including SSRs, ISSRs, AFLPs, RAPD, and SNPs (Tsumura et al., 2012). SNPs are single nucleotide polymorphisms that occur in the DNA sequence (Taheri et al., 2018), and are the most abundant and stable marker of nucleotide variation in a genome. This means that the density of SNP markers is much higher than that of any other molecular markers (Melegh et al., 2017; Rahmatalla et al., 2017). Specific-locus amplified fragment sequencing (SLAF-seq) is able to generate large datasets of SNPs (Sun et al., 2013), and has greater power than previous techniques to elucidate the genetic structure of plant populations (Narum et al., 2013). Accuracy of genotyping is ensured through deep sequencing, and costs are reduced and marker efficiency improved through the use of a pre-designed reduced representation strategy. A double barcode system is used for large populations (Sun et al., 2013). In herbaceous species, particularly those species that have experienced significant contractions in their available habitat following glacial cycling, neutral processes including changes in effective population size and allopatric divergence are expected to be of particular importance in driving population structure (Maggs et al., 2008). However, loci associated with environmental variables have been found in many studies (Yoder et al., 2014), which suggests that non-neutral processes may also have affected the observed patterns of genetic diversity.

In this study, we developed genome-wide SNP markers using SLAF-seq (Specific-Locus Amplified Fragment Sequencing) technology for G. soja populations, with 147 individuals from China, Korea and Japan. Genetic diversity, population structure, and gene flow were estimated using the newly developed genome-wide SNPs. Our research provides a valuable resource for further genome-wide association studies of G. soja and will provide guidance for the formulation of conservation strategies for this important species.

Materials and Methods

Plant materials, preparation of DNA and construction of SLAF library, and high-throughput sequencing

Leaf samples were taken from 12 Chinese populations, two Japanese populations and two Korean populations of G. soja, which together cover almost all of the main distribution area of the species. All 16 populations sampled lie outside of reserves or conservation areas, and the sample collection met the requirements of the local government in each area. Within each population, plants were randomly sampled at a minimum distance of 15 m from each other, in order to avoid the collection of ramets of the same clone. All samples were collected directly from the wild. Field experiments were approved by the National Natural Science Foundation of China (Project number: 31500459). Young, healthy leaves were collected from individuals and were dried immediately in silica gel. Between nine and ten individuals were collected from each population, with the exception of population DQ3, from which only three individuals were collected (Table 1). All samples collected were used in the subsequent analyses. Herbarium specimens of each of our samples were deposited in the biological specimen bank of the College of Horticulture and Landscape, Yunnan Agricultural University, under the voucher numbers “YNAUGLYCINE001-147”. Total genomic DNA was extracted from each sample following the cetyltrimethyl ammonium bromide (CTAB) method, using the modifications suggested by Porebski, Bailey & Baum (1997). The concentration and quality of the resulting DNA were examined with electrophoresis on a 1% agarose gel and with spectrophotometry on an ND-2000 (NanoDrop, Wilmington, DE, USA).

Table 1 Location and habitat of populations of G. soja sampled.

Population	Location	Altitude (m)	Latitude (°)	Longitude (°)	Location	
WH	Huhan, Hubei province	301	30.533	114.445	Wet land	
WN	Weinan, Shanxi province	379	34.453	109.520	Along road	
HH	Huaihua, Hunan province	890	27.715	110.81	Along road	
YW	Yiwu, Zhejiang province	72	29.338	120.038	Wet land	
NJ	Nanjing, Jiangsu province	18	32.065	118.814	Beside lake	
JN	Jinan, Shandong province	29	34.646	116.867	Barren mountain	
TJ	Tianjin, Hebei province	8	39.080	117.010	Barren land	
SY	Shenyang, Liaoning province	57	41.758	123.386	Along road	
CC	Changchun, Jilin province	125	43.871	125.241	Aside field	
HEB	Haebin, Heilongjiang province	137	45.784	126.564	Beside river	
DQ	DQ, Heilongjiang province	132	46.526	125.15	Wet land	
QQHE	QQHE, Heilongjiang province	137	47.285	123.968	Beside field	
JK	Kanagawa, Japan	12	34.959	137.139	Wet land	
JT	Tokyo, Japan	35	34.828	135.770	Wet land	
KO	Gangwon-do, South Korea	520	37.588	128.409	Wet Land	
KR	Gangwon-do, South, Korea	340	37.913	128.499	Wet land	

Specific-locus amplified fragment sequencing (SLAF-seq) is an efficient method of large-scale genotyping, and is based on a reduced representation library (RRL) and high-throughput sequencing. We used a modified SLAF-seq strategy in our experiment, with fragment sizes (including adaptors and indexes) ranging from 364 to 444 bp. The DNA was then cleaned and digested into fragments using the enzymes RsaI+HaeIII (NEB, Ipswich, MA, USA) which have been previously applied to G. max (Sun et al., 2013). Considering the close phylogenetic relationship of G. max and G. soja, we used the same enzymes to perform our SLAF pre-design experiment. The enzymes and sizes of restriction fragments were then evaluated using training data. In order to improve the efficiency of the SLAF-seq, we use training data to evaluated the enzymes and sizes of restriction fragment follow two criteria: The SLAFs should be evenly distributed through the sequences, and repeated SLAFs must be avoided. The genomic DNA from each qualifying sample was digested separately. In our study, digestion efficiency of “RsaI+HaeIII” reached 82.94%, which is within the ideal range. A single nucleotide (A) was added to the 3′ end of each of the obtained SLAF fragments, and the fragments were then connected to the Dual-Index sequencing joint. Fragments were amplified using PCR, were purified, and target fragment sizes were selected using gel tI if they passed the library quality inspection. High-throughput sequencing was performed on an Illumina HiSeqTM-2500 platform (Illumina, Inc., San Diego, CA, USA) at the Biomarker Technologies Corporation in Beijing.

Sequencing data grouping, genotyping, and genetic diversity analysis

In order to reconstruct the loci, the raw data were analyzed using the Stacks1. 0 pipeline (Catchen et al., 2011, 2013b). Data were sorted and demultiplexed according to sample barcodes using process_radtags. Raw, low-quality reads (phred score ≤ 10) were discarded and the reads were filtered to remove adapter contamination. The program ustacks (stack depth parameter (−m) = 5; a mismatch parameter (−M) = 2, maximum stacks per locus = 3) was then used to group the sample data into loci. The locus data were then merged into a catalog in cstacks. The alleles in each sample were determined by comparing the loci from each sample to the catalog in sstacks.

Species level genetic diversity in G. soja was assessed using the program populations, with all 147 samples treated together as a single population. A locus was required to be present in at least 67% of all samples in order to be eligible for inclusion in this analysis. Analysis of population level genetic diversity was conducted with each collection area treated as a population and loci were required to be present in all individuals (r = 1) in at least six populations (p = 6).

Population genetic analyses and linkage disequilibrium

The populations program in Stacks was used to calculate population genetic statistics for each SNP (number of private alleles; observed heterozygoisty (HO); expected heterozygosity (HE); nucleotide diversity (π); Wright’s F statistics FIS and FST) (Frankham, Ballou & Briscoe, 2002; Catchen et al., 2013a). The inbreeding coefficient FIS was measured for each population to investigate potentially hidden population structures within each population (Wright, 1978; Hartl & Clark, 2007). We calculated the average FST for pairwise comparisons between all sampled populations in order to investigate the genetic relatedness of the populations. The FST values were then used reconstruct a neighbor-joining tree using Mega 6.0 (Tamura et al., 2013). The correlations between genetic differentiation and geographical factors were determined using Mantel tests (Urban et al., 2002) and a matrix regression analysis (Wang, 2013) using the FST values matrices, and 10,000 permutations were used in significance testing.

Structure format files containing the SNP data were output from the populations program in Stacks to allow analysis of population level genetic structure (Pritchard, Stephens & Donnelly, 2000; Hubisz et al., 2009). Similarly, data were exported as Genepop format files to allow estimation of gene flow among populations using Genepop v4.0 (Rousset, 2008). In order to avoid tight linkage SNPs (Catchen et al., 2013b), only the first SNP at each locus was written into the Genepop and Structure files using the parameters r = 1 and p = 6. GenePop v 4.3 (Rousset, 2008) was used to test Hardy-Weinberg equilibrium (HWE) at each locus, and significance values (p) were adjusted for multiple comparisons with sequential Bonferroni correction (α = 0.05) (Rice, 1989).

The Structure files was then analyzed using program Structure 2.3 (Pritchard, Stephens & Donnelly, 2000). The initial burn-in was set to 10,000 steps and 10,000 iterations, the number of genotypic groups (K) was set to 1–20 with 10 replicates for each value. The Structure Harvester program was applied to calculate the optimal K for each analysis (Evanno, Regnaut & Goudet, 2005). In order to reveal the genetic relationships between G. soja individuals, SNPRelate was used to do the principal coordinates analysis (PCA) analysis based on the Euclidian distances of individual genotypes.

A nonlinear regression of linkage disequilibrium (LD) between polymorphic sites against distance (bp between sites) was run to estimate LD decay with physical distance. A cut-off value of r2 = 0.1 was used for the evaluation of LD decay for each population, with the r2 value for a marker distance of 0 kb assumed to be 1. Distances between the SNPs and r2 were plotted as the LD-Decay curve. r2 is usually larger where SNPs are closer, and smaller when SNPs are further far apart. The LD decay distance (LDD) is the distance during which r2 is reduced to half of its maximum value Low recombinant frequencies within a particular distance tend to result in longer LDDs while higher recombinant frequencies within the same distance result in shorter LDDs. Plink (Purcell et al., 2007) was used to calculate the LD between pairs of polymorphic sites based on the squared correlation of allele frequency.

Gene flow and migration events between populations

Maximum likelihood trees describing the historical relationships between the study populations and to infer potential migration events between them were generated in TreeMix v1.13 (Pickrell, Pritchard & Tang, 2012). TreeMix was run iteratively with the migration parameter set to −5 and the SNP block size parameter set to 10.

Results

SLAF sequencing and SNP discovery

The genome of the cultivated soybean (G. max) was used for program prediction in this project. The clean reads derived from each sample ranged between 453 and 2,202 Mb for each individual, with most reads being about 800 Mb. The average number of reads assigned to each individual was 3,859,551, with minimum and maximum read numbers per individual of 2,266,715 and 11,010,066, respectively (Table S1). Phred quality scores were high (30 ≥ 89.82%) and the GC content was found to range between 37.9% and 41.4% (Table S1). A total of 1,784,121 SLAFs were predicted, of which 548,804 were heterozygous SLAF tags. The average number of SLAF labels obtained by each individual was 202,663, with an overall average depth of 11.9×. A total of 2,436,305 SNPs were identified. SNPs which fulfilled the following criteria were then discarded: (1) those with a minor allele frequency <5%, (2) those missing more than 20% of their genotype data. Individuals missing more than 10% of the genotyped data were also discarded and (3) those SNPs which deviated from Hardy-Weinberg equilibrium (p < 0.001). A total of 56,489 SNPs were retained for downstream genetic diversity analysis. The SNPs showed a largely even distribution throughout the genome (Fig. 1).

Figure 1 Distribution of SNP labels on the chromosomes of the reference genome.

The length of chromosome is shown on the x-axis. Each bar represents a chromosome. The shade represents the SNP density for that part of the chromosome.

Genetic diversity at the species and population levels

The observed heterozygosity (Ho) was 0.0157 for all loci polymorphic at the species level, with the expected heterozygosity (He) being 0.1459, a nucleotide diversity (π) of 0.1465, and an inbreeding coefficient (FIS) of 0.8533. When considering all nucleotide positions, including the non-polymorphic ones, the observed heterozygosity decreased to 0.0004, with the expected heterozygosity decreasing to 0.0035, the nucleotide diversity decreasing to 0.0035, and the inbreeding coefficient decreasing to 0.0205 under the same conditions.

Statistical analyses for each population are given in Table 2 and Fig. 2. Across the loci that showed polymorphism in one or more populations, the average observed heterozygosity (Ho) was found to range between 0.0199 (DQ) and 0.0460 (KR), the expected heterozygosity (He) to range between 0.0119 (DQ) and 0.3492 (KR), the nucleotide diversity (π) between 0.0130 (JK) and 0.3789 (KR), and inbreeding coefficient between −0.0003 (JK) and 0.0230 (KR).

Table 2 Genetic diversity statistics for the 16 populations.

Pop ID	Private	Polymorphic loci %	Obs het	Exp het	Pi (π)	F IS	
All pos.	Variant pos.	All pos.	Variant pos.	All pos.	Variant pos.	All pos.	Variant pos.	
HH	1,755	0.30	0.0005	0.0225	0.0014	0.0557	0.0017	0.0693	0.0020	0.0829	
WH	6,398	0.54	0.0006	0.0252	0.0019	0.0799	0.0021	0.0859	0.0039	0.1595	
WN	4,317	0.43	0.0005	0.0204	0.0016	0.0674	0.0017	0.0722	0.0031	0.1286	
YW	8,434	0.82	0.0005	0.0205	0.0027	0.1106	0.0029	0.1181	0.0067	0.2774	
NJ	2,180	0.28	0.0005	0.0215	0.0011	0.0470	0.0012	0.0502	0.0014	0.0600	
JN	5,203	0.63	0.0005	0.0203	0.0022	0.0918	0.0024	0.0980	0.0049	0.2042	
TJ	10,842	0.68	0.0005	0.0204	0.0023	0.0976	0.0025	0.1038	0.0054	0.2262	
SY	4,891	0.77	0.0005	0.0203	0.0027	0.1123	0.0029	0.1193	0.0063	0.2618	
CC	1,805	0.26	0.0005	0.0206	0.0008	0.0314	0.0008	0.0334	0.0015	0.0608	
HEB	5,185	0.34	0.0005	0.0210	0.0012	0.0512	0.0013	0.0546	0.0021	0.0883	
DQ	1,793	0.18	0.0005	0.0199	0.0005	0.0199	0.0005	0.0211	0.0007	0.0277	
QQHE	12,083	0.79	0.0007	0.0293	0.0026	0.1101	0.0028	0.1173	0.0058	0.2418	
JK	2,351	0.07	0.0005	0.0200	0.0003	0.0119	0.0003	0.0130	−0.0003	−0.0126	
JT	3,356	0.26	0.0005	0.0203	0.001	0.0428	0.0011	0.0475	0.0015	0.0631	
KO	2,431	0.36	0.0005	0.0214	0.001	0.0398	0.001	0.0420	0.0022	0.0893	
KR	2,409	2.75	0.0016	0.0460	0.0121	0.3492	0.0131	0.3789	0.0230	0.6665	
Total			0.0004	0.0157	0.0035	0.1459	0.0035	0.1465	0.0205	0.8533	
Note:

Private, private allele number; Ho: observed heterozygosity; He: expected heterozygosity; π: nucleotide diversity; FIS: inbreeding coefficient of an individual relative to the subpopulation.

Figure 2 Observed genetic diversity in 16 sampled populations of wild soybean (Glycine soja).

If all nucleotides, including nonpolymorphic nucleotides were considered, the observed heterozygosity was found to lie between 0.0005 to 0.0016, with the expected heterozygosity ranging from 0.0003 to 0.0121. The observed nucleotide diversity ranged between 0.0003 and 0.0131, and the inbreeding coefficient from −0.0003 to 0.0230. The number of private alleles observed for each population ranged between 1,755 (HH) and 12,083 (QQHE). From all the measures, the highest genetic diversity was found in the KR population, followed by QQHE. The lowest nucleotide diversity and heterozygosity was seen in the JK population, with the lowest observed heterozygosity was found in the DQ population.

Population structure analysis and linkage disequilibrium

A Mantel test revealed no significant correlation between genetic distance and geographical distance (r2 = 0.0268, p = 0.104) (Fig. S1). The average pairwise FST values between different populations were used to reconstruct a neighbor-joining tree in Mega v6.0 (Fig. 3). In general, we found that individuals from the same site clustered together, however individuals from JT, SY and CC were an exception to this. Four populations from northern and central China had a close relationship (JN, TJ, WN, WH), but also clustered together with three individuals from SY (northeastern China) and six individuals from the JT population in Japan. Three populations from the Yangtze River (NJ, YW and HH) were also very similar. The two Korean populations (KO, KR) clustered together with four individuals from the Japanese JT population, and were in turn closely related to the cluster containing the northeastern Chinese populations (QQHE, DQ, CC, HEB and SY). The Japanese population JK clustered together with the populations from northern and central China (JN, TJ, WN, and WH), as did four individuals from the JT population, also from Japan. The overall trend is that G. soja populations from close to the Yangtze River have a close relationship with those from the Korean peninsula and northeastern China, even though these areas are geographically disconnected. The Japanese populations are related to those in northern China and Korea, which makes sense geographically. We found low allopatric-vicariant differentiation of these regions in our analyses.

Figure 3 Neighbor-joining tree reconstructed from clustering analysis of wild soybean accessions from 16 populations in China, Korea and Japan.

Analysis of gene flow suggested that historical migrations of G. soja may have occurred, from the south northwards across the East-Asia land-bridge. The phylogeographic history of G. soja provides us with new insights into the migration patterns of herbaceous plants across the Sino-Japanese Floristic Region.

To further investigate the population structure of the sampled G. soja populations, “admixture” and “correlated alleles frequencies” models were used to analyzed the 56,489 generated SNPs in Structure2. Changes in LnP(D) and delta K were assessed. K = 11 was best model for our data (Fig. 4A). Similar to the neighbor-joining tree, individuals within a single population were found from the posterior probabilities to have similar genetic constitution. Seven of the populations (JK, JN, TJ, WH, HEB, KO, KR, and HH) formed independent groups. WN and JN grouped together, and another group was formed by KR and four individuals from QQHE. The genetic constitutions of individuals from YW and JT were more complicated, and these samples grouped together with the populations from northeast China (DQ, CC, QQHE and SY) (Fig. 4B).

Figure 4 Inferred population structure based on 16 populations of wild soybean from China, South Korea and Japan.

(A) ADMIXTURE estimation of the number of groups for values of K ranging between 1 and 20. (B) Patterns of variation among the 147 accessions of wild soybean based on SNP analyses. The x-axis shows the different accessions. The y-axis quantify the membership probability of accessions belonging to different groups. Colors in each row represent structural components.

The principal coordinates analysis showed that individuals collected from the same site were closely related, which is consistent with the results from both the reconstructed phylogenetic tree and the Structure analysis. The PCA showed five clusters of populations. Cluster I comprised populations HH and YW, Cluster II comprised NJ and YW, and both of these two clusters contained only individuals from the Yangtze River. Cluster III comprised populations KO, KR, CC, DQ, QQHE and HEB, all of which come from northeastern China and the Korean Peninsula. Cluster IV comprised WH and JK, and cluster V comprised those populations from north China (WN, JT, SY, JN and TJ). The result is also consistent with the structure of the neighbor-joining tree (Fig. 5). Linkage disequilibrium decay curves of the 16 G. soja populations are given in Fig 6. Each colored line represents the observed LD data for a single population. A clear and rapid decline of LD is observed to occur with distance in most populations except DQ and WH, with the LD in decaying rapidly to half its initial value within about 250 kb. The r2 of populations WH and DQ tended to be stable.

Figure 5 Principal components analysis (PCA) of 16 wild soybean populations from China, South Korea and Japan, calculated using SLAF data.

Figure 6 Linkage disequilibrium (LD) decay of the G. soja genome in different populations.

The X-axis represents the distances (kb) between paired SNPs, and the Y-axis represents mean r2 of the SNP pairs within each distance region.

Genetic differentiation and gene flow among populations

Table 3 gives the calculated pairwise population Wright’s FST values for the 16 sampled G. soja populations. Genetic differentiation between populations, as calculated from the FST values, was found to be relatively high. The DQ and JK populations were the most divergent, with an FST value of 0.67, and populations YW and JN were the least divergent with a value of 0.106.

Table 3 Fst between populations collected in this study.

	HH	WH	WN	YW	NJ	JN	TJ	SY	CC	HEB	DQ	QQHE	JK	JT	KO	KR	
HH																	
WH	0.316																
WN	0.376	0.270															
YW	0.132	0.206	0.225														
NJ	0.470	0.318	0.267	0.264													
JN	0.177	0.246	0.271	0.106	0.318												
TJ	0.236	0.239	0.271	0.165	0.316	0.196											
SY	0.160	0.201	0.205	0.111	0.239	0.141	0.174										
CC	0.501	0.331	0.295	0.290	0.304	0.345	0.341	0.264									
HEB	0.410	0.340	0.388	0.246	0.465	0.293	0.256	0.256	0.494								
DQ	0.443	0.413	0.460	0.226	0.553	0.268	0.335	0.257	0.583	0.514							
QQHE	0.191	0.202	0.237	0.131	0.279	0.159	0.157	0.150	0.303	0.229	0.283						
JK	0.563	0.440	0.499	0.257	0.610	0.333	0.343	0.276	0.639	0.554	0.667	0.292					
JT	0.423	0.364	0.413	0.207	0.502	0.270	0.277	0.221	0.536	0.444	0.529	0.234	0.582				
KO	0.452	0.305	0.259	0.273	0.252	0.327	0.320	0.252	0.272	0.458	0.531	0.285	0.576	0.486			
KR	0.177	0.258	0.262	0.257	0.265	0.260	0.263	0.264	0.277	0.270	0.280	0.261	0.262	0.238	0.273		

To describe the historical relationships between these 16 sampled G. soja populations and to investigate potential migration events between them, we ran a TreeMix analysis on the 16 sampled G. soja populations. The results obtained suggest that population splits have occurred and that there has been gene flow between populations. On the TreeMix output (Fig. 7), the DQ and HEB populations cluster together as one group, and there is strong gene flow from the CC population towards QQHE. Populations TJ, JN and WN clustered together as a single group, and there was strong historical gene flow from this cluster towards the QQHE and JT populations, as well as modern gene flow from the TJ to the SY population. Overall, the general trend in gene flow was from the south towards the north, with the populations TJ, JN and WN also contributing gene flow. In summary, the general migration patterns seem to have been from the south towards the north.

Figure 7 Gene flow between wild soybean populations calculated from our SLAF data.

(A) Sample locations showing unbalanced gene flow; (B) maximum-likelihood tree. Note: The MAP is taken from CGIAR-CSI (Jarvis A., H.I. Reuter, A. Nelson, E. Guevara, 2008, Hole-filled seamless SRTM data V4, International Centre for Tropical Agriculture (CIAT), available from https://srtm.csi.cgiar.org.)

Discussion

Comparison of different molecular markers in revealing genetic diversity and differentiation in populations of G. soja

The genetic diversity and differentiation in G. soja has been investigated in the past using several different molecular markers. The diversity and structure of 11 populations of G. soja were tested by Wang & Li (2013) using nuclear microsatellite markers (SSRs), giving HO = 0.029; HE = 0.0324. Analyses of SSRs and a chloroplast locus were conducted by He et al. (2016), giving HO = 0.0324 and HE = 0.426. Zhao et al. (2006) used AFLP, ISSR and SSR data to investigate G. soja populations, with resulting in values of HE, 0.353 (AFLP), 0.226 (ISSR) and 0.157 (SSR). In the current study, we applied the high throughput sequencing technology SLAF-seq to investigate the genetic diversity of G. soja populations across the known distribution of the species. We obtained a value of HO = 0.0157 and HE = 0.1459, and different markers behaved differently in our study. Because SLAF-seq markers are genome-wide DNA tags (small fragments near specific restriction sites), they should represent the sequence characteristics of the entire genome. SLAF-seq markers are therefore believed to accurately reflect the true level of genetic diversity. Although HO and HE were different among different markers, all of the different markers show the same pattern: that the HO was much lower than HE in G. soja populations, indicating that there is a certain amount of inbreeding within the population, and that the species lacks heterozygotes. We also found that certain populations, such as KR from the Korean Peninsula, have especially high genetic diversity. This is consistent with our previous studies using SSR markers (He et al., 2012) and plastid loci (He et al., 2016). One possible reason for this high diversity could be the artificial introduction of germplasm resources from different places. Given the medicinal and scientific value of this species, more in-depth research is worth carrying out.

Levels of genetic differentiation between different G. soja populations are higher those in out-crossing species. The genetic structure within plant populations depends not only on seed and pollen dispersal distance but also on breeding type, level of self-fertilization and effective plant density (Vekemans & Hardy, 2004). In species with more restricted pollen dispersal, lower gene flow is expected to result in higher genetic differentiation, and therefore self-fertilizing species are expected to have both smaller effective populations sizes (Ingvarsson, 2002) and lower pollen movement, leading to higher genetic structure than is seen in out-crossing species (Hamrick & Godt, 1996).

When we compare our results with those from an annual, selfing plant with limited seed dispersal, the genetic differentiation is lower (Volis, Ormanbekova & Shulgina, 2016; Zhang et al., 2020). Most genetic variation occurs within rather than between populations of G. soja. While natural seed dispersal in G. soja is estimated to be less than the average of 4.5 m, possible long-distance seed dispersals of up to 200 km have been suggested on the basis of molecular data (Kuroda et al., 2006). G. soja plants are mainly self-crossing, however the large seeds have high nutritional value and are readily eaten by birds and other animals after the pods have split. G. soja also have relatively high medicinal value and have been used as a traditional Chinese medicine for many years. Human disturbance will promote seed transmission and affect the formation of patterns of genetic differentiation. This may explain the lower levels of observed genetic differentiation in G. soja populations than in some other selfing species.

Historical demography

Previous phylogeographic studies suggested that following the Quaternary glacial and inter-glacial cycles in East Asia, very few, if any, northward-southward dispersal events took place. Instead, these plant taxa survived in multiple cryptic refugia during the glaciation (Qiu et al., 2011) However, our previous SSR data and ecological niche modeling analyses (He et al., 2016), suggested that G. soja was restricted in range to southern and central China during the LGM and following the LGM the species expanded its range significantly into northern East Asia. In this study, the SLAF data suggested that gene flow between G. soja populations may have occurred across the East Asia land-bridge, which would agree with our previous findings. Gene flow was found to have occurred from the south towards the north. However, the genetic diversity index suggested that the KR and QQHE populations have high genetic diversity. This is not consistent with the idea that there was a large-scale northward range expansion in this species, because recolonized regions would be expected to show reduced genetic diversity. Therefore, it is possible that G. soja populations survived in micro-refugia in northeastern China. It has been suggested that the Changbai Mountain region suffered glaciation only above about 2,000 m during the late Pleistocene. If this is the case, the climate at lower elevations may have been mild enough during the Pleistocene glaciations that certain plant taxa could have survived in microclimatic habitats. The presence of refugia in northeastern China has been suggested by several recent phylogeographic studies (Aizawa et al., 2007; Hu et al., 2008). However, the current distribution of G. soja suggests that there may have been more than a single refuge during the glacial periods of the Pleistocene, and G. soja populations may have existed in multiple refugia, at least in the northeast of China and Korea.

Higher sea levels during and after the periods of glaciation would have meant that the CJK region was split by the East China Sea (ECS), but that there would have been a land-bridge formed by the exposed ECS basin when the sea levels decreased by c. 85–130/140 m during the glacial periods (Millien-Parra & Jaeger, 1999). Temperate deciduous forest is thought to have covered the exposed land bridge during these times (Zhang et al., 2020). The temperate flora of the area is therefore likely to have been separated and restricted to disjunct refugia during warmer times, but to have had opportunities for admixture during the glacial periods. Previous phylogeographic studies investigating Kirengeshoma (Qiu et al., 2009), Platycrater arguta (Qiu et al., 2009) and Croomia (Li et al., 2008) all suggested deep allopatric-vicariant differentiation of disjunct lineages in the CJK region (Qiu et al., 2011). In contrast with the previously studied taxa, G. soja shows lower divergence between different regions in CJK. Populations from northeastern China, southern Japan and the Korean Peninsula are genetically close. The deep allopatric-vicariant differentiation observed between the different regions of the CJK in previous phylogenetic studies and the low allopatric-vicariant differentiation we found in G. soja may result from the different habitats present in the study taxa. G. soja has a wide distribution and is sometimes able to colonize the high salt habitats along the sea shore. Because of this, G. soja might have had greater opportunity to migrate across the land-bridge and mix with other populations than did taxa with only limited distribution. Further taxa with different ranges and habits should be sampled to further investigate the biogeographical history of the CJK region. The gene flow we observed between the 16 study populations in our research provided further support for the East Asia land-bridge diffusion theory.

The Japanese populations JK and JT contained individuals from several different lineages, which suggested that these populations might have been formed from several different colonization events. We suggest that G. soja may have been introduced to Japan through long distance dispersal events mediated by migratory birds. Another possibility is that unconsidered factors, such as human-mediated dispersal or hybridization with the cultivated G. max, are influencing the population structure of the wild species. G. soja has a wide distribution across Japan, and more population sampling is necessary to resolve the phylogeographic origins of Japanese G. soja.

Implications for conservation

Two major goals in conservation include the preservation of genetic diversity and evolutionary potential and the prevention of inbreeding depression (Rauch & Bar-Yam, 2005). Currently, two main methods are used to determine populations that should receive priority protection. The first method is to use genetic variation to determine priority, but a problem with this method is that it is easy to ignore the genetic differentiation between populations, and unique alleles present in populations with low genetic variation are not effectively protected. The second method is based on genetic differentiation and considers evolutionary significant units. In this method, priority is given on the basis of the degree of genetic differentiation, that is, the more unique the population is, the more valuable it is to protect. However, it can be difficult to identify evolutionary significant units for groups with unclear pedigrees or geographical models.

Our SLAF data suggest that although G. soja resources have been seriously damaged and that a large number of populations have disappeared, G. soja retains high genetic diversity at the species level. However, some populations were found to have only very low levels of genetic diversity. For example, the nuclear diversity of the CC, DQ and JK populations was below 0.0008. In contrast, other populations were found to be highly diverse. In the KR population, for example, the nuclear diversity was 0.0131. The populations harboring high genetic diversity should be considered first in the protection of G. soja. Conservation of the original habitat, i.e., in situ protection, should be adopted for these populations.

All G. soja populations studied here could be divided into one of two lineages, and these two lineages should be treated separately when formulating protection policies. G. soja has undergone significant habitat fragmentation in recent years, and human activities have led to the extinction of the species in many areas. The wild populations comprising Lineage I were often very difficult to find, even in areas from which it had previously been reported. The genetic variation represented by various wild varieties is important for the study of the origin and evolution of the species, as well as for the breeding of cultivated varieties. However, certain varieties of G. soja, for example those with gray hairs, white flowers, and light green pods, or with yellow and brown pod have disappeared from the vast Huanghuai River basin. It is thought that land development and the construction of flood prevention dams are the reasons behind these disappearances. The collection of G. soja resources which are on the verge of extinction has therefore become urgent.

The most serious damage to Lineage II has been reported from northeastern China, in areas such as the Anbang River in Jixian County, Heilongjiang Province. In 1981, tens of thousands of square meters of G. sojas were growing along the Anbang River, but this area is now farmland, and the G. soja population has disappeared. Lack of understanding of the importance of these unique resources, indiscriminate farming practices, over-harvesting, overgrazing, as well as rural urbanization and construction of economic development zones has resulted in a nationwide decrease in G. soja numbers, and the species is now considered to be endangered. In order to actively rescue the endangered plants, the establishment of a “G. soja original habitat nature reserve” is necessary, so that this important plant can continue to have ecological and social benefits.

Certain areas have begun to realize the importance of G. soja. In 2005, the Wuqing District of Tianjin City was listed as a G. soja original habitat protection site and was officially included in the national “protection circle”. Furthermore, in 2005, experts from the Chinese Academy of Agricultural Sciences (CAAS) discovered a natural population of G. soja plants covering an area of about 3,000 m2 in Tahe County. This area was designated as a “G. soja original habitat nature reserve” by the environmental protection department. However, original habitat nature reserves are insufficient for the complete protection of G. soja, and the protection of the species needs to be strengthened.

Supplemental Information

Supplemental Information 1 IBD test.

Click here for additional data file.

Supplemental Information 2 Information for the raw and quality filtered sequence reads obtained by SLAF-seq of 147 wild soybean individuals.

Click here for additional data file.

The authors thank Okada Hiroshi and Naoko Ishikawa for the collection of the Japanese samples and Chunghee Lee for the collection of the Korean samples.

Additional Information and Declarations

Competing Interests

Author Contributions

Field Study Permissions

Data Availability

The authors declare that they have no competing interests.

Jing Meng conceived and designed the experiments, performed the experiments, authored or reviewed drafts of the article, and approved the final draft.

Guoqian Yang analyzed the data, prepared figures and/or tables, authored or reviewed drafts of the article, and approved the final draft.

Xuejiao Li performed the experiments, analyzed the data, prepared figures and/or tables, and approved the final draft.

Yan Zhao analyzed the data, prepared figures and/or tables, and approved the final draft.

Shuilian He conceived and designed the experiments, analyzed the data, authored or reviewed drafts of the article, and approved the final draft.

The following information was supplied relating to field study approvals (i.e., approving body and any reference numbers):

Field experiments were approved by the National Natural Science Foundation of China (Project numbers: 31500459).

The following information was supplied regarding data availability:

The sequencing data generated in this study for the 147 samples are available at the NCBI Sequence Read Archive: PRJNA798174; SRR17650031 to SRR17650177.

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
