# Peer review of "Population structure of wild soybean (Glycine soja) based on SLAF-seq have implications for its conservation"

_PeerJ, doi:10.7717/peerj.16415_

## Round 0.1 · original submission · Major Revisions

The authors need to include all suggestions given by reviewers.

·

Basic reporting

In this study, authors have studied population structure of 147 wild soybean accessions collected from 16 locations from China, Korea and Japan using SLAF-seq. They found that the KR population has highest genetic diversity. The general migration patterns might have occurred from the south towards the north, and while population can be divided into two lineages and may have a different protection policies.

The manuscript is well organized and written with sufficient details. However, I did not find the BioProject accession PRJNA798174 and associated runs with accession numbers from SRR17650031 to 17650177 on NCBI SRA. Authors should submit the data and provide link in the revised version of the manuscript.

Experimental design

Please show variance explained by PC1 and PC2 on figure 5.

Please explain why only two clusters or lineages inferred from this figure when there are multiple group exists. I would say at least four prominent groups.

Please add confidence values (such as bootstrap) for divergence point in the neighbor-joining tree. At present, there is no way to understand what is the accuracy of the branch divergence and weights.

Validity of the findings

Major comments

1. Korean soybean population shows exceptionally high diversity. Authors should explain possible reasons for such a high diversity and should contextualize findings with published reports.

2. Figure 5 and 6 shows that accessions have four different trends yet authors described only two lineages based on figure 5 and do not properly explained reasons behind their choices. Please explain the results from these figures in details and again contextualize findings with published reports.

Reviewer 2 ·

Basic reporting

The manuscript

Population structure of wild soybean (Glycine soja) based on SLAF-seq have implications for its conservation
is an interesting study. However I would suggest to make following changes before final publication.
Comment 1:
The abstract provides a good summary of the study; however, it would benefit from mentioning the key findings more explicitly.
Comment 2:
The sentence at the end of the introduction, "Samples were analyzed using SLAF-seq (Specific-Locus Amplified Fragment Sequencing) to investigated: the genetic diversity and population genetics of G. soja as well as any possible implications for the conservation of this species," is incomplete and lacks proper structure. It should be revised for clarity and completeness.
Comment 3:
Grammar and Spelling: In the introduction, there are a few minor grammatical issues and typos:
• In line 37, "Fujita et al. 1997)," has an extra comma.
• In line 38, "thought" should be "though."
• In line 40, "mean" should be "The mean."
Comment 4:
Ensure consistent terminology throughout the introduction. For instance, "Glycine soja" and "wild soybean" are used interchangeably. It's essential to select one term and stick with it for consistency.
Comment 5:
Consider including a brief transition sentence at the end of the introduction that introduces the methods section, setting up the reader's expectations for the next part of the paper.

Experimental design

Comment 6:
Explain why SLAF-seq was chosen as the method for this study. What advantages does it offer over previous techniques like isozymes, DNA loci, and SSRs in investigating the genetic diversity of Glycine soja populations?
Comment 7:
The introduction briefly mentions that Glycine soja is experiencing population declines across its natural range due to habitat loss and fragmentation. It would be beneficial to emphasize the urgency of these conservation issues and highlight why understanding genetic diversity is crucial for addressing them.
Comment 8:
The introduction could better emphasize the connection between understanding the genetic diversity and population structure of Glycine soja and its implications for conservation practices. Highlight why this research is not only scientifically valuable but also important for the species' preservation.
Comment 9:
Specify the reason for selecting the 12 Chinese populations, two Japanese populations, and two Korean populations. Was there a specific sampling strategy in place?
Comment 10:
Clarify if all 147 samples were included in subsequent analyses, and if not, provide the criteria for sample inclusion or exclusion.
Comment 11:
Explain the significance of the minimum 15m separation between sampled individuals.
Comment 12:
Mention whether any permits or ethical considerations were involved in collecting plant samples from the wild, as such details can be crucial in research ethics.
Comment 13:
DNA Extraction: While the methodology mentions the use of the CTAB method for DNA extraction, it would be beneficial to include more details or references for readers who may want to replicate the procedure. Additionally, specify if any modifications were made to the standard CTAB method.
Comment 14:
Clarify why fragment sizes ranging from 364 bp to 444 bp were chosen for the SLAF-seq strategy. Provide a brief explanation of the significance of this range.

Validity of the findings

Comment 15:
Explain the rationale behind using the enzymes RsaI+HaeIII for DNA digestion. Elaborate on why these enzymes were chosen and what role they played in the experiment.

Library Quality Inspection: Mention the specific criteria or standards used for the library quality inspection. What parameters were considered to determine whether a fragment passed the inspection?

---

## Round 0.2 · accepted · Accept

The revised manuscript is now scientifically sound to proceed further. The manuscript is acceptable for publication.

·

Basic reporting

The authors have adequately addressed all the concerns in the revised manuscript, and I no longer have any reservations.

Experimental design

no comment

Validity of the findings

no comment

Additional comments

no comment

Reviewer 2 ·

Basic reporting

The suggested changes have been made.

Experimental design

the suggested changes in experimental design have been made.

Validity of the findings

The suggested changes have been made.

Additional comments

The manuscript is revised thoroughly as per suggested.